# Predictors of positive tuberculin skin test in neonates exposed to pulmonary tuberculosis

Yun Choi[1][☯], In Kyoung Kim[1][☯], So Jung Kim[1], Hye Sung Kim[1], Young Ae Kang[2], Jin Su Song[3,4]*

1 Division of Infectious Disease Response, Capital Regional Center for Disease Control and Prevention, Korea Disease Control and Prevention Agency, Seoul, Republic of Korea, 2 Department of Internal Medicine, Division of Pulmonary and Critical Care Medicine, Severance Hospital, Yonsei University College of Medicine, Seoul, Republic of Korea, 3 Graduate School of Global Development and Entrepreneurship, Handong Global University, Pohang, Republic of Korea, 4 Seoul National University College of Medicine, Seoul, Republic of Korea

☯ These authors contributed equally to this work.
* dicaful@snu.ac.kr, dicaful@gmail.com

**Data Availability Statement:** Data cannot be shared publicly because of protecting personal information by the Korea Disease Control and Prevention Agency (KDCA). You may contact the Division of Research Support, Korea National

## Abstract

### Background

Neonates are at risk of nosocomial tuberculosis (TB) infection from health care workers (HCWs) in neonatal care facilities, which can progress to severe TB diseases. Tuberculin skin test (TST) is commonly used for TB diagnosis, but its accuracy in neonates is influenced by various factors, including bacilli Calmette-Guérin (BCG) vaccination. This study aimed to identify predictors of positive TSTs in neonates exposed to HCWs with pulmonary TB.

### Methods

A retrospective observational study was conducted to compare the frequency of predictors between TST-positive and TST-negative neonates. Demographic, epidemiological, and clinical data of neonates exposed to TB, along with that of HCW and household contacts, were collected retrospectively through contact investigations with the Korean National TB Surveillance System (KNTSS) database. TSTs using 2 tuberculin units of purified protein derivative RT23 were performed on exposed neonates at the end of preventive TB treatment. Firth logistic regression was performed to identify predictors of TST positivity.

### Results

Contact investigations revealed that 152 neonates and 54 HCWs were exposed to infectious TB index cases in 3 neonatal care facilities. Of 152 exposed neonates, 8 (5.3%) had positive TST results. Age of 6 days or more at the initial exposure is a statistically significant predictor of positive TST (Firth coefficient 2.1, 95% confidence interval 0.3–3.9, $P = 0.024$); BCG vaccination showed no statistical significance in both univariable and multivariable analysis. Sex, prematurity, exposure duration, duration from initial exposure to contact investigation, and isoniazid preventive treatment duration were not significant predictors.

Institute of Health (KNIH), KDCA (Tel: +82-43-719-7364, email contact point: suheecho8610@korea.kr) for requesting the detailed data in the study.

**Funding:** The authors received no specific funding for this work.

**Competing interests:** The authors have declared that no competing interests exist.

## Conclusion

Age at the initial exposure is a significant predictor of positive TST in neonates exposed to active pulmonary TB. Given the complexities of TST interpretation, including false positives due to BCG vaccination, careful risk assessment is necessary for appropriate decision-making and resource allocation in the management of neonatal TB exposure.

## Introduction

South Korea is an intermediate tuberculosis (TB) burden country, with the annual incidence of TB of 44 persons per 100,000 population in 2021 [1]. TB remains a significant public health issue in South Korea. Neonates, in particular, are at risk of nosocomial TB infection from health care workers (HCWs) in postpartum care centers, hospitals, and intensive care units [2–4], having the potential to develop the most severe forms of TB such as TB meningitis or disseminated TB disease. Early TB detection and treatment in neonates, therefore, are crucial to ensure their survival and prevent long-term complications [5].

Tuberculin skin test (TST) is a widely used diagnostic tool for TB infection due to its low cost, simplicity, and feasibility. However, confirming TB infection in young children can be challenging due to various factors affecting the test results. One major factor is the bacilli Calmette-Guérin (BCG) vaccination, which reduces the specificity of the TST test and can lead to unnecessary treatment for those without TB Infection [6–9]. Some studies, however, presented conflicting results, indicating that BCG vaccination has only a limited effect on TST among infants [10,11].

The purpose of this study was to identify predictors of a positive TST in neonates exposed to active pulmonary TB. Given the low risk of TB transmission in postpartum or neonatal care centers and intensive care units in South Korea and elsewhere [12–14], we hypothesized that some positive TST results could be due to the BCG vaccination effect rather than recent exposure and infection.

## Methods

### Study design

Epidemiologic investigations were conducted following 3 TB exposure events in different neonatal care facilities. Details of the demographic, epidemiologic, and clinical data of exposed neonates, HCWs, and household contacts were collected retrospectively from investigation reports and the Korean National TB Surveillance System (KNTSS) database. We identified TST-positive and TST-negative neonates and compared the frequency of a risk factor in each group to determine the association between the predictors and TST positivity.

### Setting and index cases

According to the Korea Disease Control and Prevention Agency (KDCA) guidelines [15], we investigated 3 neonatal care facilities where an employee had been diagnosed with pulmonary TB. Given an index case's clinical and diagnostic characteristics, the infectious period was determined to begin 4 weeks before the diagnostic test of the index case. The neonatal care rooms of 2 postpartum care centers (Facility A and B) and an obstetrics hospital (Facility C) have 19, 14, and 30 newborn cribs, respectively. Facility A had a higher room density of 0.7 person/m$^2$ compared to Facility B and C. During the exposure period, heating, ventilation and

**Table 1. Characteristics of index cases, contacts for investigation, and exposed environment in 3 neonatal care facilities.**

| Characteristics | Facility A | Facility B | Facility C |
|---|---|---|---|
| Index case | | | |
| Demographic and clinical feature | | | |
| Sex/Age | female/58 | female/60 | female/46 |
| Occupation | nursing assistant | nursing assistant | nursing assistant |
| Underlying disease | none | hypertension/DM | none |
| Previous TB history | none | none | none |
| Previous LTBI test | negative | negative | unknown |
| Symptom | none | none | none |
| Cavity on CXR or CT | none | none | none |
| Diagnostic test result | | | |
| Sputum AFB smear | negative | negative | negative |
| TB-PCR | positive | negative | negative |
| Xpert MTB/RIF | MTB detected RIF sensitive | MTB not detected | MTB detected RIF sensitive |
| Culture (solid or liquid) | negative | positive (solid) | positive (liquid) |
| Resistance to TB drugs | none | none | none |
| Contacts for investigation (n) | | | |
| Neonate | 44 | 22 | 86 |
| Health care worker | 28 | 13 | 13 |
| Household | 3 | 3 | 1 |
| The environment of the exposed place | | | |
| Mechanical air ventilation | In operation | In operation | In operation |
| Portable air cleaner | In operation | In operation | In operation |
| Size of neonatal care room (m$^2$) | 33 | 36.4 | 82.6 |
| Density (person/m$^2$) | 0.7 | 0.2 | 0.2 |

Abbreviations: DM, diabetes mellitus; MTB, *mycobacterium tuberculosis*; TB, tuberculosis; CXR, chest X-ray; CT, computed tomography; AFB, acid-fast bacilli; PCR, polymerase chain reaction; RIF, rifampicin.

air conditioning (HVAC) systems in all 3 facilities were operational, but their effectiveness such as air changes per hour and carbon dioxide level was not measured. Each neonatal care room in the facilities was equipped with portable air cleaners (Table 1). Facility A's neonatal care room had windows that occupied 75% of the wall area, providing natural ventilation and ample light. In contrast, Facility C's room had windows covering only 25% of the wall area; Facility B had no windows. Strict infection prevention and control protocols were enforced in 3 facilities for coronavirus disease 2019 (COVID-19). At Facility A and B, visitors other than the mother's spouse were prohibited; visitors to Facility C were only permitted to observe neonates through the care room window twice a day, for one hour each time. An index case at each facility is a nursing assistant who worked 8-hour shifts, 4 to 5 days a week. Their duties included caring for newborns in a neonatal care unit, which involved feeding, soothing, and changing diapers for approximately 5 to 6 hours per day. Most of the mothers fed their newborns during the day. A mother and her newborn were allowed to spend two hours together each day in a mother's room with natural ventilation in the neonatal care room of Facility A during this time. Index cases A and B were primarily involved in newborn care, while index case C was primarily responsible for medical charting. On average, each index patient cared for approximately 4 newborns. Given the worker's typical work pattern, we estimate they had about 30–40 minutes of daily close contact with each newborn. The index cases had casual

contact with other nursing staff in the dressing room or during duty handover that lasted for approximately 30 minutes.

### Index case A

During the annual TB examination on August 9, 2021, an asymptomatic 58-year-old nursing assistant was found to have a nodule in the right lung, but no cavities on chest X-ray (CXR). At the physician's discretion, a chest computed tomography (CT) scan was performed for an accurate diagnosis which confirmed the earlier findings, suggesting pulmonary tuberculosis. Because of the scarcity of sputum, fiberoptic bronchoscopy was performed for the diagnosis. Her bronchoalveolar lavage (BAL) fluid showed negative results for acid-fast bacilli (AFB) and *Mycobacterium tuberculosis* cultures. However, polymerase chain reaction (PCR) and Xpert MTB/RIF (Cepheid, Sunnyvale, CA, USA) were positive for *M. tuberculosis* without rifampin resistance. In 2017, the interferon-γ release assay (IGRA) was negative and the CXR during the 2020 recruitment check-up was normal.

### Index case B

A 60-year-old nursing assistant without symptoms was found to have abnormal findings on CT during a routine check-up on August 10, 2021. Repeated CT and CXR showed a low attenuation mass in the right apex with no cavities. The AFB smears, TB-PCR, Xpert MTB/RIF assay, and liquid cultures returned negative. However, the solid cultures from the BAL fluid and lung biopsy were positive. In March 2021, her recruitment check-up showed no abnormalities in the CXR.

### Index case C

A 46-year-old nursing assistant had a CXR on an annual medical examination on December 11, 2021, revealing abnormal findings but no cavity. CT showed conglomerated nodules with tree-in-bud appearances in the right upper lung. Sputum smears for AFB and TB-PCR were negative, but Xpert MTB/RIF assay and culture in liquid media were positive. She was asymptomatic but had a calcified lesion in 2019 and 2020, regarded as spontaneously healed TB.

### Management of exposed neonates, HCWs, and households

The exposed neonates were given CXRs and isoniazid prophylaxis (10 mg/kg/day) for 3 months. TSTs using 2 Tuberculin Units (TU) of purified protein derivative (PPD) RT23 (Statens Serum Institut, Copenhagen, Denmark) were performed using the Mantoux method at the end of preventive TB treatment. A positive TST is indicated by an induration of ≥5 mm for neonates without BCG vaccination and ≥10 mm for those with BCG vaccination. HCWs and household contacts were initially offered CXRs and IGRAs. After 8–12 weeks, they underwent the second CXRs, and those with negative results at the initial IGRA were offered repeat IGRAs (Fig 1). The KNTSS was used with epidemiological investigation results to verify previous TB history, latent tuberculosis infection (LTBI) test results, and treatment outcomes. All contacts were followed up for about 18 months after the last exposure with the index cases until June 30, 2023.

### Statistical analysis

Descriptive statistics were used to summarize the demographic and clinical characteristics of the study population. Continuous variables were presented as mean with standard deviation (SD) or medians with interquartile range (IQR); categorical variables as frequencies with percentages. We conducted a comparative analysis of variables using the $\chi^2$ test, Mann–Whitney

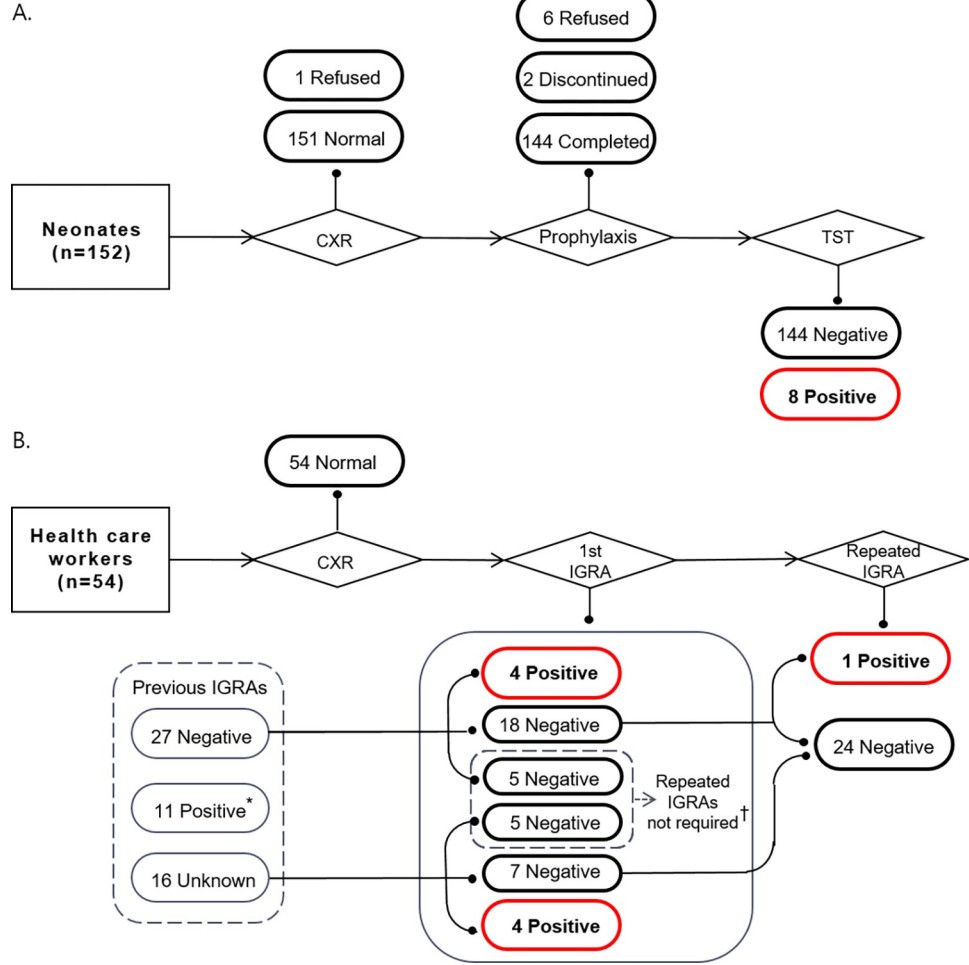

**Fig 1.** Investigation flow chart and outcomes of (A) neonates and (B) health care workers exposed to an index case with pulmonary tuberculosis. * Seven had completed latent TB treatment. 2 health care workers were registered in the system by verbally stating that they had completed latent TB or spinal TB treatment, while 1 health care worker was left untreated. The results of 1 health care worker are unknown † Ten health care workers with negative results in the first IGRA were excluded from repeated IGRAs as the initial IGRA was provided 8 weeks after the end of the exposure Abbreviations: CXR, chest X-ray; TST, tuberculin skin test; IGRA, interferon-γ release assay.

*U* test, or Kruskal–Wallis test based on the nature of each variable. To identify the association with TST positivity, Firth logistic regression was utilized to reduce biases caused by rare events [16]. We included factors with a *P* value less than 0.10 from univariable analysis in multivariable logistic regression. All statistical analyses were conducted using IBM SPSS Statistics ver. 23.0 (IBM Corp., Armonk, N.Y., USA), except for Firth logistic regression performed with STATA software ver. 14.0 (Stata Corp, College Station, TX).

## Ethical approval

This study was reviewed by the Institutional Review Board (IRB) of the KDCA and was exempt from review (KDCA-2023-06-03-PE-01). Written informed consent was waived because the patients' records and information were anonymized and de-identified prior to the analysis. The need for informed consent was waived by the IRB of the KDCA. Data were assessed from June 8, 2023 to June 30, 2023 for research purposes.

## Results

### Outcomes of neonatal investigation

Of the 152 exposed neonates, 72 (47.4%) were male. The median chronological and corrected ages of neonates at the start of the investigation were 21 (IQR 11–31) days old and 11 (IQR 1–26) days old, respectively. The median duration of exposure to an index was 5 (IQR 3–6) days, with facility A having a longer duration of 7 (IQR 5–8) days: neonates at facility A tended to stay longer. Facility B started contact investigation earlier than Facility A and C, but delayed CXR screening and isoniazid preventive therapy (IPT) until solid culture results were available. There were no significant differences in prematurity, sex, and TST results across 3 facilities (Table 2).

**Table 2. Characteristics and TST results of the 152 exposed neonates in 3 neonatal care facilities.**

| Characteristics | | Total | | Facility | | $\chi^2$, Z | $p^*$ |
| --- | --- | --- | --- | --- | --- | --- | --- |
| | | | A | B | C | | |
| Age at the start of the investigation, days, median (IQR) | | 25 (15–36) | 32 (22–41) | 24 (14–36) | 20 (13–32) | 14.5 | <0.001 |
| Chronological, median (IQR) | | 21 (11–31) | 30 (20–39) | 17 (7–29) | 16 (9–28) | 23.4 | <0.001 |
| Corrected, median (IQR) (n = 130) | | 11 (1–26) | 19 (7–33) | NA | 8 (0–21) | -3.4 | <0.001 |
| Prematurity (n = 130) | | | | | | | |
| | <37 weeks | 11 (8.5) | 6 (13.6) | NA | 5 (5.8) | 2.3 | 0.129 |
| | ≥37 weeks | 119 (91.5) | 38 (86.4) | NA | 81 (94.2) | | |
| Sex | | | | | | | |
| | Male | 72 (47.4) | 22 (50.0) | 12 (54.5) | 38 (44.2) | 0.9 | 0.629 |
| | Female | 80 (52.6) | 22 (50.0) | 10 (45.5) | 48 (55.8) | | |
| Delivery type (n = 130) | | | | | | | |
| | Normal delivery | 50 (38.5) | 23 (52.3) | NA | 27 (31.4) | 5.4 | 0.021 |
| | Cesarean delivery | 80 (61.5) | 21 (47.7) | NA | 59 (68.6) | | |
| BCG vaccination | | | | | | | |
| | Unvaccinated | 60 (39.5) | 13 (29.5) | 0 (0.0) | 47 (54.7) | 24.4 | <0.001 |
| | Vaccinated | 92 (60.5) | 31 (70.5) | 22 (100.0) | 39 (45.3) | | |
| BCG type[†] (n = 92) | | | | | | | |
| | Intradermal | 53 (57.6) | 19 (61.3) | 17 (77.3) | 17 (43.6) | 6.8 | 0.033 |
| | Percutaneous | 39 (42.4) | 12 (38.7) | 5 (22.7) | 22 (56.4) | | |
| Exposure duration, days, median (IQR) | | 5 (3–6) | 7 (5–8) | 6 (4–7) | 4 (3–5) | 38.6 | <0.001 |
| Age at the days of TST, median (IQR) | | 92 (91–96) | 92 (90–96) | 97 (92–102) | 92 (91–95) | 7.4 | 0.024 |
| TST results | | | | | | | |
| | Positive | 8 (5.3) | 5 (11.4) | 0 (0.0) | 3 (3.5) | 5.0 | 0.080 |
| | Negative | 144 (94.7) | 39 (88.6) | 22 (100) | 83 (96.5) | | |
| TST induration | | | | | | | |
| | Positive, mm, median (IQR) | 12 (10–13) | 12 (10–12) | 0 (0.0) | 14 (11–14) | -1.4 | 0.167 |
| | Negative, mm, median (IQR) | 0 (0–4) | 0 (0–5) | 5 (1–8) | 0 (0–0) | 20.6 | <0.001 |
| Initial exposure to contact investigation, days, median (IQR) | | 23 (13–31) | 27 (18–32) | 18 (7–31) | 19 (12–31) | 7.9 | 0.019 |
| Initial exposure to contact screening with chest X-ray, days, median (IQR) | | 32 (21–39) | 33 (24–35) | 58 (47–70) | 26 (18–38) | 55.0 | <0.001 |
| Initial exposure to initiation of IPT, days, median (IQR) (n = 145) | | 31 (21–39) | 34 (25–37) | 55 (46–68) | 26 (17–38) | 46.7 | <0.001 |

* Depending on the variables' characteristics, one of the following tests was used: Kruskal–Wallis test, Mann–Whitney $U$ test, or $\chi^2$ test.

[†] In South Korea, newborns are vaccinated with either intradermal BCG Danish (strain 1331; Statens Serum Institut) or percutaneous BCG Tokyo (strain 172; Japan BCG Laboratory).

Abbreviations: NA, not available; IQR, interquartile range; BCG, bacilli Calmett-Guérin; TST, tuberculin skin test; IPT, isoniazid preventive therapy.

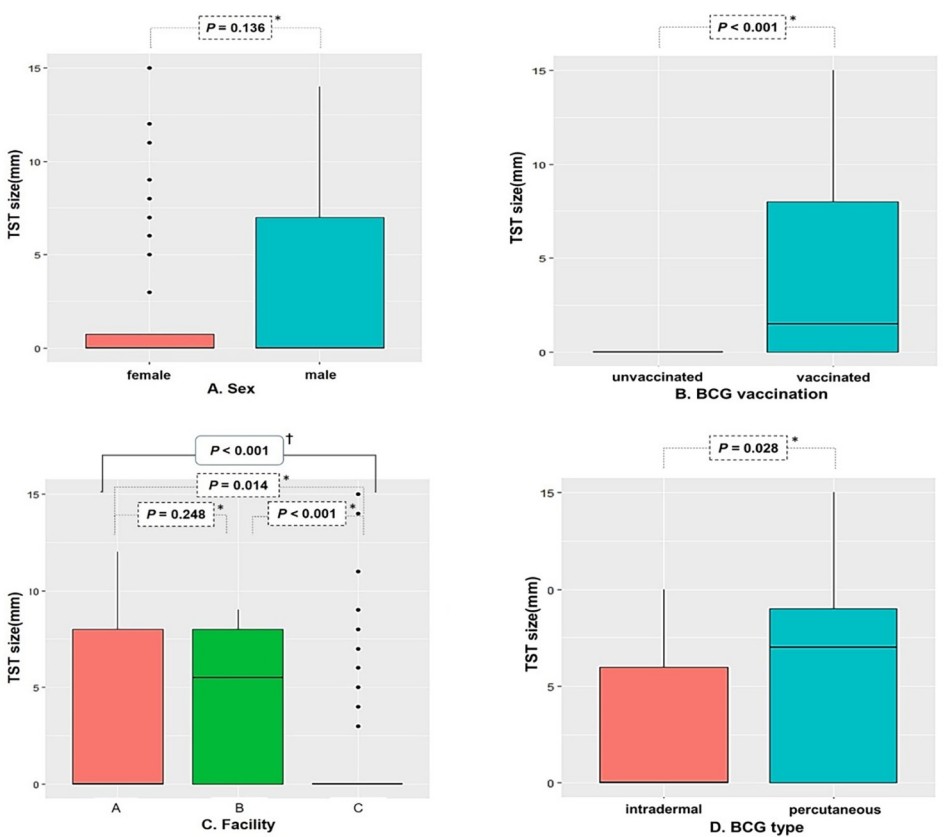

**Fig 2.** Comparison of TST induration size (mm) according to (A) sex (n = 152), (B) BCG vaccination (n = 152), (C) facility (n = 152), and (D) BCG type (n = 92). *Mann–Whitney *U* test †Kruskal–Wallis test.

Ninety-two (60.5%) neonates received BCG vaccination: 53 (57.6%) intradermally (BCG Danish strain 1331; Statens Serum Institut) and 39 (42.4%) percutaneously (BCG Tokyo strain 172; Japan BCG Laboratory). 60 (39.5%) did not receive vaccination. Out of the 151 neonates who underwent CXRs, all except one refuser showed no indications of active pulmonary TB in their results. Of the 152 neonates, except 6 refusers and 2 drop-outs, 144 have completed 3 months of IPT. At the 3-month TST screening, 8 infants, comprising 5 males and 3 females, tested positive (Fig 1A). Neonates not given BCG vaccinations showed no skin reaction, with an induration size of 0 mm. On the other hand, neonates who received BCG vaccinations had varying sizes of induration (range 0–15). Compared with the intradermal route the percutaneous administration of BCG vaccine resulted in a higher skin test reaction (mean ± SD 5.2 ± 5.0 vs. 3.0 ± 3.6) (Fig 2D).

## Outcomes of HCWs and household investigation

Of the 54 HCWs, 45 (83.3%) were nurses while the remaining comprised 4 nutritionists or cooks, 1 cleaner, and 4 administrative staff. Thirty-eight had previous IGRA results from 2015 to 2021, and 16 had no results recorded in the KNTSS. In accordance with KDCA guidelines [15], 11 HCWs were excluded from further LTBI tests as they had previously tested positive in IGRAs. None of the 54 HCWs had abnormalities at the baseline or second CXRs. Eight (18.6%) of the 43 HCWs with previous IGRAs were positive in the first IGRA. One HCW with negative in the initial IGRA was positive in a repeated test (Fig 1B). All 7 family members did

not exhibit any TB signs on CXRs. Although they initially tested negative on the first IGRAs, one member tested positive on a subsequent IGRA.

### Treatment results of contacts diagnosed with LTBI

Of the 18 contacts diagnosed with LTBI, 17 (94.4%) received prophylactic treatment, and 16 completed the LTBI treatment. Eight infants started LTBI treatment: 3 infants with 3-month rifampin plus isoniazid, and 5 with 9-month isoniazid completed their treatment. Out of 9 HCWs subject to LTBI treatment, 8, except one refuser, initiated treatment with 3-month rifampicin plus isoniazid (n = 6) and 4-month rifampicin (n = 2). Of these, 7 completed their regimen and 1 discontinued due to elevated liver enzymes. A family member with LTBI was initiated with 4-month rifampicin and completed the regimen. No cases of active TB have been identified during the observation period (S1 Table).

### Predictors associated with a positive TST among neonates exposed to TB index cases

During the investigation period, 8 (5.3%) of the 152 neonates exposed to a pulmonary TB patient had positive TST results. In the univariable analysis, age of 6 days or more at the day of initial exposure to an index case (Firth coefficient [coef.] 2.8, 95% confidence interval [CI] 1.0–4.6, $P$ = 0.002) and age of 45 days or more at the start of IPT (Firth coef. 2.1, 95% CI 0.6–3.6, $P$ = 0.006) are significant predictors associated with a positive TST; BCG vaccination is a not a statistically significant predictor (Firth coef. 2.5, 95% CI -0.4–5.4, $P$ = 0.088). In the multivariable analysis that excluded the age at the start of IPT due to the issue of multicollinearity, the age of 6 days or more at the initial exposure is the only statistically significant predictor associated with positive TST (Firth coef. 2.1, 95% CI 0.3–3.9, $P$ = 0.024). On the other hand, sex, prematurity, exposure duration, duration from initial exposure to contact investigation, and IPT duration were not found to be significant predictors of TST positivity in the univariable and multivariable models (Table 3).

In sensitivity analysis using a 5 mm cut-off point in BCG-vaccinated neonates, all of the 42 neonates who had TST-positive had received BCG vaccination, while of the 110 TST-negatives, 50 (45.4%) had received BCG vaccination, and 60 (54.6%) had not. The Haldane-Anscombe correction indicates that BCG vaccination is a statistically significant factor that influenced TST positivity (odds ratio [OR] 101.8, 95% CI 6.1–1696.6, $P$ = 0.001).

### Discussion

Our research indicated that being aged 6 days or more at the initial exposure was associated with positive TST in multivariable regression analysis; BCG vaccination showed no statistical significance both in univariable and multivariable analysis. In sensitivity analysis with a 5-mm cutoff point among the BCG-vaccinated, however, BCG vaccination was a determining factor to predict a TST positivity with a statistical significance. To the best of our knowledge, this is the first study to find the predictors of positive TSTs among neonates exposed to an infectious index in a congregate setting.

It was an unexpected finding that a relatively older age on the day of initial exposure or at the start of IPT was associated with positive TSTs. It is postulated that TST results may be falsely negative in a younger age group due to their less developed cellular immunity [17]. Or greater protection against infections may have occurred at an earlier stage in life due to vertical or innate immunity acquired from the mother [18,19]. However, it is still unclear at what age this becomes a factor. It is also likely that a newborn less than a week old was better protected from TB transmission by nursing staff, who handled them less while feeding, caring, or soothing in a shielded crib.

**Table 3. Predictors associated with a positive tuberculin skin test among neonates exposed to an index case with pulmonary TB.**

| | | TST | | χ2 | p* | Univariable | | | | Multivariable | | | |
|---|---|---|---|---|---|---|---|---|---|---|---|---|---|
| | | Negative n (%) | Positive n (%) | | | Firth coef. | 95% CI | | p† | Firth coef. | 95% CI | | p† |
| Age at the day of initial exposure to an index | | | | | | | | | | | | | |
| | <6 days | 110 (99.1) | 1 (0.9) | 15.7 | <0.001 | ref | | | | ref | | | |
| | ≥6 days | 34 (82.9) | 7 (17.1) | | | 2.8 | 1.0 | 4.6 | 0.002 | 2.1 | 0.3 | 3.9 | 0.024 |
| Age at the start of IPT‡ (n = 137) | | | | | | | | | | | | | |
| | <45 days | 105 (98.1) | 2 (1.9) | 10.4 | <0.001 | ref | | | | | | | |
| | ≥45 days | 32 (84.2) | 6 (15.8) | | | 2.1 | 0.6 | 3.6 | 0.006 | | | | |
| Sex | | | | | | | | | | | | | |
| | Male | 67 (93.1) | 5 (6.9) | 0.8 | 0.379 | ref | | | | | | | |
| | Female | 77 (96.3) | 3 (3.8) | | | -0.6 | -2.0 | 0.8 | 0.402 | | | | |
| Prematurity (n = 130) | | | | | | | | | | | | | |
| | <37 weeks | 10 (90.9) | 1 (9.1) | 0.2 | 0.672 | ref | | | | | | | |
| | ≥37 weeks | 112 (94.1) | 7 (5.9) | | | -0.8 | -2.6 | 1.1 | 0.423 | | | | |
| Exposure duration | | | | | | | | | | | | | |
| | <6 days | 97 (93.3) | 7 (4.6) | 1.4 | 0.233 | ref | | | | | | | |
| | ≥6 days | 47 (97.9) | 1 (2.1) | | | -0.9 | -2.7 | 0.9 | 0.329 | | | | |
| Age at the days of TST | | | | | | | | | | | | | |
| | <96 days | 104 (95.4) | 5 (4.6) | 0.3 | 0.552 | ref | | | | | | | |
| | ≥96 days | 40 (93.0) | 3 (7.0) | | | 0.5 | -0.9 | 1.9 | 0.484 | | | | |
| Duration from initial exposure to contact investigation | | | | | | | | | | | | | |
| | <94 days | 91 (92.9) | 7 (7.1) | 1.9 | 0.162 | ref | | | | | | | |
| | ≥94 days | 53 (98.1) | 1 (1.9) | | | -1.1 | -2.9 | 0.7 | 0.239 | | | | |
| IPT period | | | | | | | | | | | | | |
| | <68 days | 100 (92.6) | 8 (7.4) | 2.90 | 0.089 | ref | | | | | | | |
| | ≥68 days | 37 (100.0) | 0 (0.0) | | | -1.8 | -4.7 | 1.0 | 0.208 | | | | |
| BCG vaccination | | | | | | | | | | | | | |
| | Unvaccinated | 60 (100.0) | 0 (0.0) | 5.51 | 0.019 | ref | | | | ref | | | |
| | Vaccinated | 84 (91.3) | 8 (8.7) | | | 2.5 | -0.4 | 5.4 | 0.088 | 1.4 | -1.7 | 4.5 | 0.384 |

* χ² test.

† Firth logistic regression was used in univariable and multivariable regression analysis.

‡ Excluded in multivariable regression analysis due to multicollinearity with age at the day of initial exposure.

Abbreviations: CI, confidence interval; Coef., coefficient; TST, tuberculin skin test; IPT, isoniazid prevention therapy; BCG, bacilli Calmett-Guérin.

Despite no significant result in univariable and multivariable analysis, the correlation between BCG vaccination and TST induration size was demonstrated in box plot analysis (Fig 2B). Likewise, two meta-analyses showed that vaccination with BCG significantly increases the likelihood of a positive TST [20,21]. In contrast, a cross-sectional study in Brazil has shown that BCG had a low impact on TST in children <2 years old without apparent contact with TB patients [10], and a cohort study in India demonstrated that no infant who received BCG vaccination within the first month of life had a TST reaction size greater than 10 mm [11]. Two studies, however, differ from ours in vaccine strain, administration method, age group tested, environmental mycobacteria, and sample size, thus leading to the challenges of interpretation.

Notably, we did not find an association between exposure duration and TST positivity among exposed neonates. The lack of association is in contrast to previous studies that have reported

higher TST positivity rates and TB diseases in those who had more prolonged contact with an active TB patient [2,14]. Several factors contribute to the lack of association: the low infectivity of index cases, protected environment in baby cribs with minimum exposure, enhanced infection prevention measures due to COVID-19 control, and prompt contact investigation and IPT initiation. Given the distribution of induration size, the age when vaccinated, and the interval between vaccination and testing, along with the results of predictor analysis, TST positivity in the present study is deemed more closely related to the effect of BCG vaccination rather than TB infection.

In the present study, induration diameters in infants vaccinated with the percutaneous route were larger than those with intradermal vaccination (Fig 2D), which aligns with other studies [22–25]. However, there were conflicting reports that showed stronger skin reactions in intradermal vaccination than in percutaneous injection [26,27]. Additionally, the immunogenicity and protective efficacy against TB by the routes of administration have not been clearly determined. Therefore, it is important to consider the route of administration as a potential confounder when interpreting the results of the TST, conducting risk assessment, evaluating clinical efficacy, and determining cost-effectiveness within and between studies.

Two index cases in our study were detected through mandatory employment examination, which has been in effect since February 2016 under the Korean Tuberculosis Prevention Act [28]. According to the amended law, the heads of medical institutions, postnatal care business entities, kindergartens, childcare centers, and schools are obliged to conduct TB and LTBI examinations for the employees. Individual TB and LTBI screening can promptly identify employees with TB infection, minimizing transmission risk in high-risk congregate settings such as neonatal care facilities. The benefits of mandatory employment TB examination vary based on local epidemiology and contextual factors. As illustrated in the study, however, regular screening of HCWs in an intermediate or high-burden country can play a vital role in identifying infected employees, preventing onward transmission, and possibly saving costs.

Our study has some acknowledged limitations. First, there were only 8 TST-positive neonates with a zero count of TST-positives among the unvaccinated which can lead to biased or inflated parameter estimates, although we utilized Firth logistic regression to minimize small-sample and rare event bias by penalizing odds ratios. Therefore, with the need for a cautious interpretation of our findings, further study with an adequate sample size needs to be followed up to revalidate our findings. Second, induration sizes were measured by multiple individuals in referral hospitals, potentially causing inter-observer variation. Finally, our study's findings should be applied with caution to other healthcare settings due to varying transmission risks based on factors such as exposure duration or intensity, infectiousness of index case, implementation of infection control measures, and TB prevalence or incidence in the country.

In conclusion, our study reveals that age at the initial exposure is a statistically significant predictor of positive TST in neonates exposed to HCWs with pulmonary TB; BCG vaccination appears to be associated with positive TST. However, other risk factors, such as prematurity, sex, and exposure duration, and BCG vaccination were not associated with positive TSTs. Given the uncertainties and complexities of TST interpretation in neonates, including a high false positive rate due to the effect of BCG vaccination, careful risk assessment—source case, exposure environment, and susceptibility and potential severity of disease—is warranted to lead to appropriate resource allocation, precautionary measures, and response.

## Supporting information

**S1 Table. Treatment results of contacts exposed to an index case with pulmonary tuberculosis.** * 1 HCW with a previous TB history was excluded [†] 1 HCW refused treatment [‡] 1 HCW stopped treatment due to side effect Abbreviations: IQR = interquartile range; HCW = health

care worker; LTBI = latent tuberculosis infection; RFP = rifampicin; INH = isoniazid.
(PDF)

## Acknowledgments

We thank public health officers for case investigations and medical professionals in referral hospitals for their contributions to patient management.

## Author Contributions

**Conceptualization:** Yun Choi, In Kyoung Kim, Young Ae Kang, Jin Su Song.

**Data curation:** Yun Choi, In Kyoung Kim.

**Investigation:** Yun Choi, In Kyoung Kim, So Jung Kim, Hye Sung Kim.

**Methodology:** Young Ae Kang.

**Project administration:** Jin Su Song.

**Supervision:** Jin Su Song.

**Validation:** Young Ae Kang, Jin Su Song.

**Visualization:** Yun Choi, In Kyoung Kim.

**Writing – original draft:** Yun Choi, In Kyoung Kim, Jin Su Song.

**Writing – review & editing:** Yun Choi, In Kyoung Kim, So Jung Kim, Hye Sung Kim, Young Ae Kang, Jin Su Song.

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
