## [Decision Letter · Decision Letter 0]

30 Oct 2023

PONE-D-23-28374Predictors of positive tuberculin skin test in neonates exposed to pulmonary tuberculosisPLOS ONE

Dear Dr. Song,

Thank you for submitting your manuscript to PLOS ONE. After careful consideration, we feel that it has merit but does not fully meet PLOS ONE’s publication criteria as it currently stands. Therefore, we invite you to submit a revised version of the manuscript that addresses the points raised during the review process.

We look forward to receiving your revised manuscript.

Kind regards,

Rebecca F. Baggaley

Academic Editor

PLOS ONE

Journal Requirements:

Reviewers' comments:

Reviewer's Responses to Questions

**Comments to the Author**

1. Is the manuscript technically sound, and do the data support the conclusions?

Reviewer #1: Partly

Reviewer #2: Yes

2. Has the statistical analysis been performed appropriately and rigorously? 

Reviewer #1: No

Reviewer #2: Yes

3. Have the authors made all data underlying the findings in their manuscript fully available?

Reviewer #1: Yes

Reviewer #2: Yes

4. Is the manuscript presented in an intelligible fashion and written in standard English?

Reviewer #1: Yes

Reviewer #2: Yes

5. Review Comments to the Author

Reviewer #1: In this article Song et. al., have assessed the TST positivity predictors in neonates exposed to TB index cases (healthcare workers) and undergone isoniazid prophylaxis regimen. As claimed by the author, this could become the first published study to find the predictors of positive TSTs among neonates exposed to an infectious index. The identification and characterization of the index cases and follow up success rate of this study should be appreciated. However, the study needs further clarity and major revision to justify the conclusion drawn from it.

All the study subjects were exposed to index cases as defined in this paper. However, looking at the features of index cases (as in Table 1) and with the preventive measure taken for Covid-19 (Page 4, line 83), the likelihood of exposure is minimal. This has also been raised by the authors in the discussion (page 19, line 262). The authors should elaborate on the nature and likelihood of exposure.

In the study, the percentage of TST positive cases among exposed indexes are 5.3%. If the objective was to investigate the impact of potential exposure in neonates on TST, then a cohort of neonates with no-exposure to any index cases will be useful for comparision.

The primary outcome of this study is- age (>< 6 days) at the initial exposure a predictor of TST reactivity. The chi-square test shows BCG vaccination as a determinants of TST positivity (univariate Firth coefficient test also showed some trend (0.08)). Also the Haldane-Anscombe correction emphasized on BCG vaccination as a critical factor (Page 15, line 222). Since the number of TST positive subjects among non-vaccinated cases is zero, it may over-fit statistical models used. The authos should elaborate on the BCG vaccination regimen used for the study subjects. When were they vaccinated after birth? Was there any heterogeneity in the time duration between vaccination and TST test?

Page 6, Table 1: Xpert MTB/RIF positive gives an impression of being Rif-resistant.

Page 11, Table-1: Please clarify the missing 3% in prematurity [<37 weeks: 11 (5.4%), >37 weeks: 119 (91.6)] and 14.5% in delivery type.

Page 16, Table-3: Whether all the subjects completed the 3 months IPT regimen. What was the rationale behind setting the 68 days cut-off?

Page 16, Table-3: Percentage distribution <6 days 110 (99%) 1 (1%) and >6 days 34 (83%) 7 (17%) will give better impression on the data.

Reviewer #2: Author has attempted to see the predictors of positive TST in neonates exposed to Pulmonary Tuberculosis. Study is effectively planned and implemented. However, following points need clarifications:

1. Here index cases of health care workers have been detected by chest CT only. Is it routine practice to do TB screening with CT chest in author's setting? If no, author can explain what is routine practice and why CT were done in these cases.

2. In line no. 103, in index case A, how the nodule was detected? Is it on chest x-ray or CT scan?

3. In line no. 105, for the same index case A, for which sample PCR or GeneXpert was done ? Is it sputum or biopsy sample?

4. Line no. 175, numbers are not matching. 152- 6 refusers - 2 refusers= 144. But author has written 142 completed the treatment.

5. Line No. 18, the author has commented that treatment was given to 18 contacts. But there is no mention of those 11 contacts that were detected positive, first time by IGRA. What happened to those on follow-up? Were they also put on preventive treatment?

6. Lines no. 220 and 236; in sensitivity analysis, author used 5 mm cut-off, but in methodology, author explained that 10mm cut off was chosen for those already taken BCG. Please explain why for analysis 5mm cut-off was considered. It may have given few false positive cases.

6. PLOS authors have the option to publish the peer review history of their article (what does this mean?). If published, this will include your full peer review and any attached files.

Reviewer #1: No

Reviewer #2: No

---

## [Author Response · Author response to Decision Letter 0]

3 Dec 2023

Reviewer 1: I have incorporated all of your concerns and suggestions into my revision. We are grateful for the insightful comments and valuable improvements to our paper. Thank you for your help.

Reviewer 2: I have incorporated all of your questions and comments into my revision. Thank you.

---

## [Decision Letter · Decision Letter 1]

31 Jan 2024

PONE-D-23-28374R1Predictors of positive tuberculin skin test in neonates exposed to pulmonary tuberculosisPLOS ONE

Dear Dr. Song,

Thank you for submitting your manuscript to PLOS ONE. After careful consideration, we feel that it has merit but does not fully meet PLOS ONE’s publication criteria as it currently stands. Therefore, we invite you to submit a revised version of the manuscript that addresses the points raised during the review process.

We look forward to receiving your revised manuscript.

Kind regards,

Rebecca F. Baggaley

Academic Editor

PLOS ONE

Journal Requirements:

Reviewers' comments:

Reviewer's Responses to Questions

**Comments to the Author**

1. If the authors have adequately addressed your comments raised in a previous round of review and you feel that this manuscript is now acceptable for publication, you may indicate that here to bypass the “Comments to the Author” section, enter your conflict of interest statement in the “Confidential to Editor” section, and submit your "Accept" recommendation.

Reviewer #3: (No Response)

2. Is the manuscript technically sound, and do the data support the conclusions?

Reviewer #3: Partly

3. Has the statistical analysis been performed appropriately and rigorously? 

Reviewer #3: Yes

4. Have the authors made all data underlying the findings in their manuscript fully available?

Reviewer #3: Yes

5. Is the manuscript presented in an intelligible fashion and written in standard English?

Reviewer #3: Yes

6. Review Comments to the Author

Reviewer #3: I appreciate the author (s) for coming up with this work on assessing the predictors of positive tuberculin skin tests in neonates exposed to pulmonary tuberculosis. It is an area with limited information and claimed by the author, it could be the first study in such a setting. We could build on this work for future studies. I am in support and acknowledged the work of other reviewers. However, the study needs to clarify minor corrections moving forward.

1.Neonates are at risk of nosocomial tuberculosis (TB) infection from health care workers (HCWs) in neonatal care facilities, which can progress to severe diseases. The statement "which can progress to severe diseases." It sounds confusing as it may mean nosocomial tuberculosis (TB) infection from health care workers (HCWs) could progress to several other severe diseases than severe TB disease or different forms of TB.

2.The statement "BCG vaccination showed marginal statistical significance (Firth coefficient 2.5, 95% CI -0.4–5.4, P = 0.088) only in univariable analysis" in the abstract and result section could be use with caution. It is important to note that marginal significance differs from true statistical significance. Utilizing the term "marginally significant" is an indirect way of admitting that the resulting p-value was not statistically significant while attempting to give it the appearance of statistical significance. Researchers commit this fraudulent act because the more significant their results are (or, in this case, seem), the more likely the study is to be published; in turn, these publishments may reward researchers with grants or financial aid to continue their research or begin new studies (Lybrand et al., 2021).

3.While I appreciate the author's outstanding work describing the study area and settings, the author did not mention the study design in the abstract and methodology section. In addition, the author needs to be clear on sampling (sample size) and procedures.

References

Lybrand, B., Blackhart, G., Parish, A., Lowe, H., 2021. Investigating the Misrepresentation of Statistical Significance in Empirical Articles.

7. PLOS authors have the option to publish the peer review history of their article (what does this mean?). If published, this will include your full peer review and any attached files.

Reviewer #3: **Yes: **Dickens Odongo, Department of Environmental Health and Disease Control, Faculty of Public Health, Lira University, Lira, Uganda.

---

## [Author Response · Author response to Decision Letter 1]

27 Feb 2024

Reviewer 3: I have incorporated all of your concerns and suggestions into my revision. 

We are grateful for the insightful comments and valuable improvements to our paper.

---

## [Decision Letter · Decision Letter 2]

19 Apr 2024

Predictors of positive tuberculin skin test in neonates exposed to pulmonary tuberculosis

PONE-D-23-28374R2

Dear Dr. Jin Su Song,

We’re pleased to inform you that your manuscript has been judged scientifically suitable for publication and will be formally accepted for publication once it meets all outstanding technical requirements.

Kind regards,

Novel N. Chegou, Ph.D

Academic Editor

PLOS ONE

Additional Editor Comments (optional):

Reviewers' comments:

Reviewer's Responses to Questions

**Comments to the Author**

1. If the authors have adequately addressed your comments raised in a previous round of review and you feel that this manuscript is now acceptable for publication, you may indicate that here to bypass the “Comments to the Author” section, enter your conflict of interest statement in the “Confidential to Editor” section, and submit your "Accept" recommendation.

Reviewer #3: All comments have been addressed

2. Is the manuscript technically sound, and do the data support the conclusions?

Reviewer #3: Yes

3. Has the statistical analysis been performed appropriately and rigorously? 

Reviewer #3: Yes

4. Have the authors made all data underlying the findings in their manuscript fully available?

Reviewer #3: Yes

5. Is the manuscript presented in an intelligible fashion and written in standard English?

Reviewer #3: Yes

6. Review Comments to the Author

Reviewer #3: (No Response)

7. PLOS authors have the option to publish the peer review history of their article (what does this mean?). If published, this will include your full peer review and any attached files.

Reviewer #3: No

---

## [Editor Report · Acceptance letter]

26 Apr 2024

PONE-D-23-28374R2 

PLOS ONE

Dear Dr. Song, 

I'm pleased to inform you that your manuscript has been deemed suitable for publication in PLOS ONE. Congratulations! Your manuscript is now being handed over to our production team.

Kind regards, 

on behalf of

Prof Novel Njweipi Chegou 

Academic Editor

PLOS ONE